# Adaptive Structural Fingerprints for Graph Attention Networks

**Kai Zhang**
Department of Computer & Information Sciences
Temple University
Philadelphia PA 19122, USA
kzhang980@gmail.com

**Yaokang Zhu & Jun Wang**[*]
School of Computer Science and Technology
East Chine Normal University, Shanghai China
52184501026@stu.ecnu.edu.cn
jwang@sei.ecnu.edu.cn

**Jie Zhang**[†]
Institute of Brain-Inspired Intelligence
Fudan University, Shanghai China
jzhang080@gmail.com

## Abstract

Graph attention network (GAT) is a promising framework to perform convolution and massage passing on graphs. Yet, how to fully exploit rich structural information in the attention mechanism remains a challenge. In the current version, GAT calculates attention scores mainly using node features and among one-hop neighbors, while increasing the attention range to higher-order neighbors can negatively affect performance, reflecting the over-smoothing risk of GAT (or graph neural networks in general), and the ineffectiveness in exploiting graph structural details. In this paper, we propose an "adaptive structural fingerprint" (ADSF) model to fully exploit graph topological details in graph attention network. The key idea is to contextualize each node with a weighted, learnable receptive field encoding rich and diverse local graph structures. By doing this, structural interactions between the nodes can be inferred accurately, thus significantly improving subsequent attention layer as well as the convergence of learning. Furthermore, our model provides a useful platform for different subspaces of node features and various scales of graph structures to "cross-talk" with each other through the learning of multi-head attention, being particularly useful in handling complex real-world data. Empirical results demonstrate the power of our approach in exploiting rich structural information in GAT and in alleviating the intrinsic oversmoothing problem in graph neural networks.

## 1 Introduction

Many real-world data set are represented naturally as graphs. For example, citation networks specify the citation links among scientific papers; social media often need to explore the significant amount of connections between users; biological processes typically involve complex interactions such as protein-protein-interaction (PPI). In these scenarios, the complex structures such as the graph topology or connectivities encode crucial domain-specific knowledge for the learning and prediction tasks. Examples include node embedding or classification, graph classification, and so on.

The complexity of graph-structured data makes it non-trivial to employ traditional convolutional neural networks (CNN's). The CNN architecture was originally designed for images whose pixels are located on a uniform grids, and so the convolutional filters can be reused everywhere without having to accommodate local structure changes (LeCun & Kavukcuoglu, 2010) . However, extending CNN to deal with arbitrary structured graphs can be non-trivial. To solve this problem, graph neural networks (GNN) were early proposed by Gori et al. (2005) and Sperduti (1997), which adopt

---

[*]Corresponding author
[†]Corresponding author

an iterative process to propagate the state of each node, followed by a neural network module to generate the output until an equilibrium state is reached. Recent development of GNN can be categorized into spectral and non-spectral approaches. Spectral approaches employ the tools in signal processing and transform the convolutional operation in the graph domain to much simpler operations of the Laplacian spectrum (Bruna et al., 2014), and various approaches have been proposed to localize the convolution in either the graph or spectral domain (Henaff et al., 2015; Defferrard et al., 2016; Kipf & Welling, 2017). Non-spectral approaches define convolutions directly on the neighboring nodes. As a result, varying node structures have to be accommodated through various processing steps such as fixed-neighborhood size sampling (Hamilton et al., 2017), neighborhood normalization (Niepert et al., 2016), or learning a weight matrix for each node degree (Duvenaud et al., 2015) or neighborhood size (Hamilton et al., 2017). Recently, residual network is also introduced to graph neural networks (Zhang & Meng, 2019).

Recently, graph attention network (GAT) proves a promising framework by combining graph neural networks with attention mechanism (Velickovic et al., 2017). The attention mechanism allows dealing with variable sized input while focusing on the most relevant parts, and has been widely used in sequence modelling (Bahdanau et al., 2015), machine translation (Luong et al., 2015), and visual processing (Xu et al., 2015). The GAT model further introduces attention module into graphs, where the hidden representation of the nodes are computed by repeatedly attending over their neighbors' features, and the weighting coefficients are calculated inductively based on a self-attention strategy.

Despite the numerous success, how to exploit structural information in GAT remains an challenge. Note that attention scores in GAT are computed mainly based on the content of the nodes; the structures of the graph are simply used to mask the attention, e.g., only one-hop neighbors will be attended. When considering attention among higher order neighbors, however, the performance of GAT deteriorates (see experimental section for details). This is closely related to over-smoothing of GNN's (Li et al., 2018) and reflects the weakness of GAT in effectively exploiting graph structural information. In this paper, we believe that the topology or "shapes" of local edge connections scan provide a valuable guidance on how to exploit rich structural information of graphs. For example, in social networks or biological networks, a community or pathway may be composed of nodes that are densely inter-connected with each other but several hops away. Therefore, it can be beneficial attend neighbors from the same community, even they show no direct connections. To achieve this, simply checking k-hop neighbors would seem insufficient and a thorough exploration of structural landscapes of the graph becomes necessary.

In order to fully exploit rich, high-order structural details in graph attention networks, we propose a new model called "adaptive structural fingerprints". The key idea is to contextualize each node within a local receptive field composed of its high-order neighbors. Each node in the neighborhood will be assigned a non-negative, closed-form weighting based on local information propagation procedures, and so the domain (or shape) of the receptive field will adapt automatically to local graph structures and the learning task. We call this weighted, tunable receptive field for each node its "structural fingerprint". The structural fingerprint encodes important structural details and will be used in conjunction with the node feature to compute an improved attention layer. Furthermore, our approach provides a useful platform for different subspaces of the node features and various scales of local graph structures to coordinate with each other in learning multi-head attention, being particularly beneficial in handling complex real-world graph data sets.

In Section 2, we introduce the proposed method, including limitation of content-based attention, construction of adaptive structural fingerprints, and the algorithm workflow. In Section 3, we discuss related work. Section 4 reports empirical evaluations and the last section concludes the paper.

## 2 Exploiting Graph Structural Details in Attention

### 2.1 Limitations of content-based attention

The "closeness" between two nodes should be determined from both their content and structure. Here we illustrate the importance of detailed graph structures in determining node similarities. In Figure 1(a), suppose the feature similarity of node-pairs (A,B) and (A,C) are similar. Namely, content-based attention will be similar for this two node pairs. However, from structural point of view, the attention between (A,B) should be much stronger than that for (A,C). This is because A

and B are located in a small, densely inter-connected community, and they share a significant portion of common neighbors; while node A and C does not have any common neighbor. In Figure 1(b), node A and node B are not direct neighbors, and connecting them takes three edges. As a result, their features will not directly affect each other in the message passing of GAT[1]. However, both A and B are strongly connected to a dense community, and node B further connects to the community hub. Therefore, it is reasonable for A and B to directly affect each other.

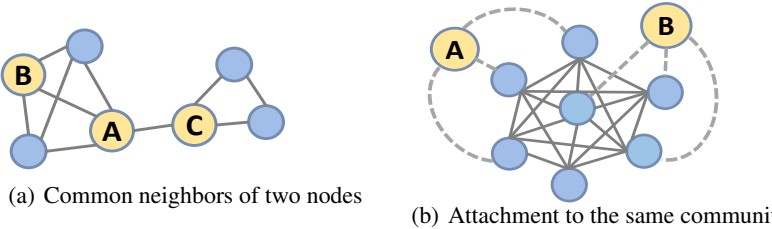

(a) Common neighbors of two nodes

(b) Attachment to the same community

Figure 1: Detailed graph structures provide important guidance in determining node attention.

In these examples, feature based similarity alone is insufficient in computing a faithful attention. One may need to take into account structural details of higher-order neighbors and how they interact with each other. In the literature, structural clues have long been exploited in solving problems of clustering, semi-supervised learning, and community detection . For example, normalized cut minimizes edge connections between clusters for node partitioning (Shi & Malik, 2000); mean-shift uses density peaks to identify clusters (Comaniciu & Meer, 2002); low-density separation (Chapelle & Zien, 2005) assumes that class boundaries pass through low-density regions, leading to successful semi-supervised classification; in community detection, densely connected subgraphs are the main indicator of community (Girvan & Newman, 2002).

In the following, we show how to build "adaptive structural fingerprint" to extract informative structural clues to improve graph attention and henceforth node embedding and classification.

## 2.2 ADAPTIVE STRUCTURAL FINGERPRINTS

Key to exploiting the structural information is the construction of the so-called "adaptive structural fingerprints", by placing each node in the context of the its local "receptive field". Figure 2 is an illustration. For any node $i$, consider a spanning process that locates a local subgraph around $i$, for example, all the nodes within the $k$-hop neighbors of $i$. Denote the resultant subgraph as $(V_i, E_i)$, where $V_i$ and $E_i$ are the set of nodes (dark red) and edges (black) in this neighborhood. In the meantime, each node in $i$ will be assigned a non-negative weight, denoted by $\mathbf{w}_i \in \mathbb{R}^{n_i \times 1}$, which specifies the importance of each node in shaping the receptive filed also determines the effective "shape" and size of the field. The structural fingerprint of node $i$ will be defined as $F_i = (V_i, \mathbf{w}_i)$.

## 2.3 CONSTRUCTION OF THE STRUCTURAL FINGERPRINTS

Intuitively, the weight of the nodes should decay with their distance from the center of the fingerprint. A simple idea is to compute the weight of the node $j$ using a Gaussian function of its distance from the center node $i$, i.e., $\mathbf{w}_i(j) = \exp(-\frac{dis(i,j)^2}{2h^2})$, which we call Gaussian decay. Alternatively, we can map the node distance levels $[1, 2, ..., k]$ to weight levels $\mathbf{u} = [u_1, u_2, ..., u_k]$ which are non-negative and monotonic; we call nonparametric decay, which is more flexible and will be used in our experiments. In both cases, the decay parameter (Gaussian bandwidth $h$ or nonparametric decay profile $\mathbf{u}$) can be optimized, making the structural fingerprint more adaptive to the learning process.

In practice, it is more desirable if node weights can be automatically adjusted by local graph structures (shapes or density of connections). To achieve this, we propose to use Random Walk with Restart (RWR). Random walks were first developed to explore global topology of a network by simulating a particle iteratively moving among neighboring nodes (Lovasz, 1993). If the particle is

---

[1]If one considers multiple message passing steps, higher-order neighbors may finally affect each other; but this can be difficult to analyze and convergence can be slower than interactions within one step.

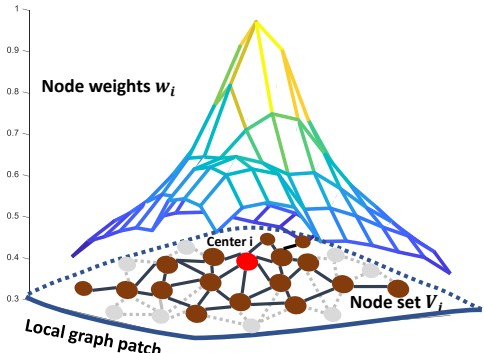

Figure 2: Structural fingerprint for a given node $i$, denoted by $F_i = (V_i, \mathbf{w}_i)$, where $V_i$ is the set of nodes in this local receptive field and $\mathbf{w}_i$ is the contributing weights of the nodes.

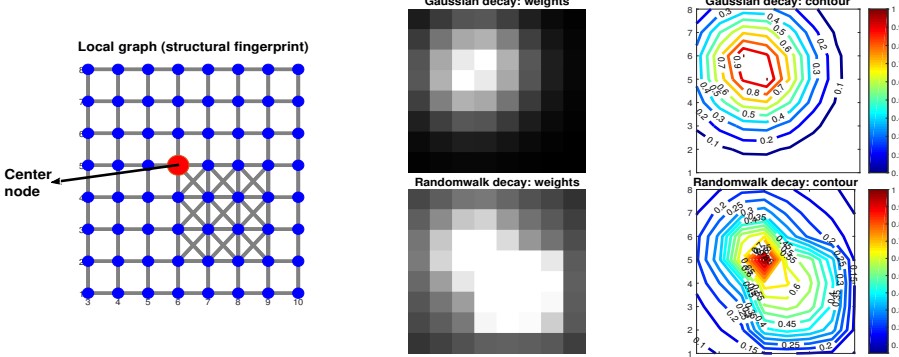

Figure 3: Weights $\mathbf{w}_i$ of a toy structural fingerprint. Left: the local subgraph/fingerprint; middle: visualization of the weights; right: contours of the weights. Gaussian decay and RWR(random walk with restart) decay leads to different weight contours, the latter more adaptively adjusting the weights to structural details of the local graph.

forced to always restart in the same node (or set of "seed" nodes), random walk with restart (RWR) can then quantify the structural proximity between the seed(s) and all the other nodes in the graph, which has been widely used in information retrieval (Tong et al., 2006; Pan et al., 2004).

Consider a random walk with restart on a structural fingerprint centered around node $i$, with altogether $n_i$ nodes and adjacency matrix $\mathbf{E}_i$. The particle starts from the center node $i$ and randomly walks to its neighbors in $V_i$ with a probability proportional to edge weights. In each step, it also has a certain probability to return to the center node. The iteration can be written as

$$\mathbf{w}_i^{(t+1)} = c \cdot \tilde{\mathbf{E}}_i \mathbf{w}_i^{(t)} + (1 - c) \cdot \mathbf{e}_i$$

where $\tilde{\mathbf{E}}$ is the transition probability matrix by normalizing columns of $\mathbf{E}_i$, $c \in [0, 1]$ is a tradeoff parameter between random walk and restart, and $\mathbf{e}_i$ is a vector of all zeros except the entry corresponding to the center node $i$. The converged solution can be written in closed form as

$$\mathbf{w}_i = (\mathbf{I} - c \cdot \tilde{\mathbf{E}}_i)^{-1} \mathbf{e}_i. \tag{1}$$

The $\mathbf{w}_i$ quantifies the proximity between the center node $i$ and all other nodes of the fingerprint, and in the meantime naturally reflects local structural details of the graph. The $c$ controls the decaying rate (effective size) of the fingerprint: if $c = 0$, $\mathbf{w}_i$ will all be zeros except the $i$th node; if $c = 1$, $\mathbf{w}_i$ will be the stationary distribution of a standard random walk on graph $(V_i, \mathbf{E}_i)$. In practice, $c$ will be optimized so that the fingerprint adapts naturally to both the graph structure and the learning task.

In Figure 3, we illustrate Gaussian-based and RWR-based fingerprint weights. We created a toy example of local receptive field, which for convenience is a uniform grid but with a small denser "patch" on the right bottom. As can be expected, the Gaussian based weight decay has contours that are center-symmetric. In comparison, the RWR automatically takes into account salient local structures and so the contours will be biased towards the dense subgraph. This is particularly desirable in case the center node is close to (or residing in) a community; it will then be represented more closely by the nodes from community, achieving the effect of a "structural attractor".

As can be seen, the "structural attractor" effect in building the fingerprint with RWR coincides well with commonly used clustering assumptions, namely, the structural fingerprint of a node will emphasize more on densely inter-connected neighbors within predefined range. Since highly weighted nodes are more likely to come from the same cluster as the center node, the fingerprint of each node is supposed to provide highly informative guidance on its structural identity, thus improving evaluation of node similarites and finally the graph attention.

## 2.4 ALGORITHM DESCRIPTION

In this section we will exploit both the content and structural details of the graph in the GAT framework (Velickovic et al., 2017). Our algorithm is illustrated in Figure 4.

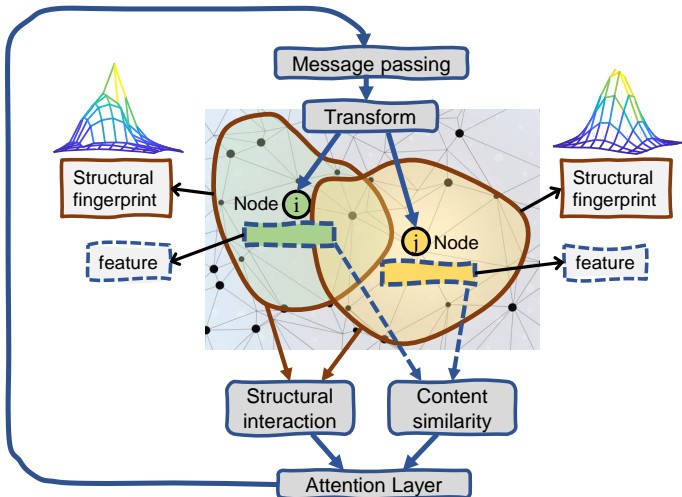

Figure 4: The work flow of adaptive structural fingerprint model.

Suppose we want to compute attention coefficients between a pair of nodes $i$ and $j$, each with their features and structural fingerprints. Content-wise, features of the two nodes will be used to compute their content similarity; structure-wise, structural fingerprints of the two nodes will be used to evaluate their interaction. Both scores will be incorporated in the attention layer, which will then be used in the message passing step to update node features. Following Velickovic et al. (2017), we also apply a transform on the features, and apply multiple steps of message passing.

More specifically, given a graph of $n$ nodes $G = (V, \mathbf{E})$ where $V$ is the set of the nodes and $\mathbf{E}$ is the set of the edges; let $\{\mathbf{h}_i\}_{i=1}^n$ be the $d$-dimensional input features for each node. We will follow the basic structure of GAT algorithm (Velickovic et al., 2017) and describe our algorithms as follows.

- Step 1. Evaluate the content similarity between node $i$ and $j$ as

$$e_{ij} = \mathcal{A}_{fea}(\mathbf{W}\mathbf{h}_i, \mathbf{W}\mathbf{h}_j) \qquad (2)$$

where $\mathbf{W} \in \mathbb{R}^{d \times d}$ is the transformation that maps the node features to a latent space, and function $\mathcal{A}_{fea}(\cdot, \cdot)$ computes similarity (or interaction) between two feature $\mathbf{h}_i$ and $\mathbf{h}_j$,

$$\mathcal{A}_{fea}(\mathbf{W}\mathbf{h}_i, \mathbf{W}\mathbf{h}_j) = \mathbf{a}^\top(\mathbf{W}\mathbf{h}_i || \mathbf{W}\mathbf{h}_j) \qquad (3)$$

- Step 2. Evaluate structural interaction between the structural fingerprints of node $i$ and $j$,

$$s_{ij} = \mathcal{A}_{str}(F_i, F_j) \tag{4}$$

where $\mathcal{A}_{str}(F_i, F_j)$ quantifies the interaction between two fingerprints. Let $\mathbf{w}_i$ and $\mathbf{w}_j$ be the node weights of the fingerprints for node $i$ and $j$, as discussed in Section 2.3. Then we can adopt the weighted Jacard similarity to evaluate the structural interactions, as

$$\mathcal{A}_{str}(F_i, F_j) = \frac{\sum_{p \in (V_i \cup V_j)} \min(w_{ip}, w_{jp})}{\sum_{p \in (V_i \cup V_j)} \max(w_{ip}, w_{jp})}$$

Here, with an abuse of notations, we have expanded $\mathbf{w}_i$ and $\mathbf{w}_j$ to all the nodes in $V_i \cup V_i$ by filling zeros. We can consider the following smooth version, or other smooth alternative[2].

$$\max(x, y) = \lim_t \frac{\log(e^{t \cdot x} + e^{t \cdot y})}{t}, \quad \min(x, y) = \lim_t -\frac{\log(e^{t \cdot (-x)} + e^{t \cdot (-y)})}{t}$$

- Step 3. Normalize (sparsify) feature similarities (2) and the structural interactions as (4)

$$\bar{e}_{ij} \leftarrow \frac{\exp(\text{LeakyRelu}(e_{ij}))}{\sum \exp(\text{LeakyRelu}(e_{ik}))}, \quad \bar{s}_{ij} \leftarrow \frac{\exp(s_{ij})}{\sum \exp(s_{ik})} \tag{5}$$

and then combine them to compute the final attention

$$a_{ij} = \frac{\alpha(\bar{e}_{ij})\bar{e}_{ij} + \beta(\bar{s}_{ij})\bar{s}_{ij}}{\alpha_{(}\bar{e}_{ij}) + \beta(\bar{s}_{ij})}. \tag{6}$$

Here $\alpha(\cdot)$ and $\beta(\cdot)$ are transfer functions (such as Sigmoid) that adjust feature similarity and structure interaction scores before combining them. For simplicity (and in our experiments), we use scalar $\alpha$ and $\beta$, which leads to a standard weighted average.

- Step 4. Perform message passing to update the features of each node as

$$\mathbf{h}_i^{(t+1)} = \sigma \left( \sum_{j \in \mathcal{N}_i} \alpha_{ij} \mathbf{W} \mathbf{h}_j^{(t)} \right)$$

Our algorithm has a particular advantage when multi-head attention is pursued. Note that our model simultaneously calculates two attention scores: the content-based $e_{ij}$ and structure-based $s_{ij}$, and combine them together. Therefore each attention head will accommodate two sets of parameters: (1) those of the content-based attention, $\mathbf{W}$ and $\mathbf{a}$ (3), which explores the subspace of the node features, and (2) those of the structure-based attention, $c$ (1), which explores the decay rate of the structural fingerprint. As a result, by learning an optimal mixture of the two attention (6), our model provides a flexible platform for different subspaces of node features and various scales of local graph structures to "cross-talk" with each other, which can be quite useful in exploring complex real-world data.

Computationally, the local receptive field will be confined within a $k$-hop neighborhood, which we call "fingerprint size"; on the other hand, both structural attention and content-based attention will be considered only when the center node of the two receptive fields have a distance below the threshold $k'$, which we call "attention range". In experiments we simply set $k = k' = 2$, and use breadth-first-search (BFS) to localize the neighborhood. As a result, the complexity involved in structure exploration will be $O(n|\overline{\mathcal{N}_{k'}}|)$, where $|\overline{\mathcal{N}_{k'}}|$ is the averaged $k'$-hop-neighbor size.

## 3 COMPARISONS WITH RELATED WORK

Note that in graph convolutional network (Kipf & Welling, 2017), the node representation is updated by $\mathbf{h}_i^{(t+1)} = \sigma \left( \sum_{j \in \mathcal{N}_i^{\mathbf{A}}} [\mathbf{D}^{-1/2} \mathbf{A} \mathbf{D}^{-1/2}]_{ij} \mathbf{h}_j^{(t)} \mathbf{W} \right)$, where $\mathbf{A}$ is the adjacency matrix, $\mathbf{D}$ is the degree matrix, $\mathbf{h}_i$ is representation of the $i$th node and $\mathbf{W}$ an embedding matrix. Namely, the message

---

[2]$\mathcal{A}_{str}(F_i, F_j) = \frac{\sum_{p \in \Omega_{ij}}^n \gamma(w_{ip}, w_{jp})}{\sum_{p \in \Omega_{ij}} \gamma(w_{ip}, w_{jp}) + \sum_{p \in \Omega_i} w_{ip} + \sum_{p \in \Omega_j} w_{jp}}$ where $\gamma(x, y)$ denotes either the arithmetic or geometric mean of $x$ and $y$, and where $\Omega_i = V_i - V_j$, $\Omega_j = V_j - V_i$, and $\Omega_{ij} = V_i \cap V_j$

| Data | # Nodes | # Edges | # Features | # Classes | #Training | # Validation | # Testing |
|------|---------|---------|-----------|-----------|-----------|--------------|-----------|
| Cora | 2708 | 5429 | 1433 | 7 | 140 | 500 | 1000 |
| Citeseer | 3327 | 4732 | 3703 | 6 | 120 | 500 | 1000 |
| Pubmed | 19717 | 44338 | 500 | 3 | 60 | 500 | 1000 |

Table 1: Summary statistics of the benchmark graph-structured data sets used in the experiment.

passing is mainly determined by the (normalized) adjacency matrix. The GAT method (Velickovic et al., 2017) replaces the fixed adjacency matrix with an inductive, trainable attention function that relies instead on the node features within one-hop neighbors. Our approach has a notable difference. First, our message passing is determined by a mixed attention from both structure and content (6). Second, the structural component of our attention is not simply based on the graph adjacency (Kipf & Welling, 2017), or one-hop neighbor (Velickovic et al., 2017), but instead relies on a local receptive field whose "shapes" are optimized adaptively through learning (1). Furthermore, our method fully exploits structural details (e.g. density and topology of local connections). There are also a number of works that explore structures in graph classification (Lee et al., 2018; Rossi et al., 2019). There, attention is used to identify small but discriminative parts of the graph, also called "graphlets" or "motifs" (Morris et al., 2019), in order to perform classification on the graph level.

## 4 EXPERIMENTS

In this section, we report experimental results of the proposed method and state-of-the-art algorithms using graph-based benchmark data sets and transductive classification problem. Our codes can be downloaded from the anonymous Github link http://github.com/AvigdorZ.

### 4.1 EXPERIMENTAL SETTING AND RESULTS

We have reported results of the following baseline algorithms: Gaussian fields and harmonic function (Gaussian Fields) (Zhu et al., 2003), manifold regularization (Manifold Reg.) (Belkin et al., 2006); Deep Semi-supervised learning (Deep-Semi) (Weston et al., 2012); link-based classification (Link-based) Lu & Getoor. (2003); skip-gram based graph embedding (Deep-Walk) (Perozzi et al., 2014); semi-supervised learning with graph embedding (Planetoid) (Yang et al., 2016); graph convolutional networks (GCN) (Kipf & Welling, 2017); high-order chebyshev filters with GCN (Chebyshev) (Defferrard et al., 2016), and the mixture model CNN (Mixture-CNN) (Monti et al., 2016). We have selected three benchmark graph-structured data set from (Sen et al., 2008), namely Cora, Citeseer, and Pubmed. The three data sets are all citation networks. Here, each node denotes one document, and an edge will connect two nodes if there is citation link between the two document; the raw features of each document are bags-of-words representations. Each node (document) will be associated with one label, and following the transductive setting in (Velickovic et al., 2017; Yang et al., 2016) we only use 20 labeled samples for each class but with all the remaining, unlabelled data for training. We split the data set into three parts: training, validation, and testing, as shown in table 1. Algorithm performance will be evaluated on the classification precision on the test split. For algorithm using random initialization, averaged performance over 10 runs will be reported.

The network structures of our methods follow the GAT method (Velickovic et al., 2017), with the following details. Altogether two layers of message passing are adopted. In the first layer, one transformation matrix $\mathbf{W} \in \mathbb{R}^{d \times 8}$ is learned for each of altogether 8 attention heads; in the second layer, a transformation matrix $\mathbf{W} \in \mathbb{R}^{64 \times C}$ is used on the concatenated features (from the 8 attention head from the first layer), and one attention head is adopted followed by a softmax operator, where $C$ is the number of classes. The number of parameters is $64(d + C)$. For the Pubmed data set, 8 attention heads are used in the second layer due to the larger graph size. Adam SGD is used for optimization, with learning rate $\lambda = 5 \times 1e - 4$. See more details in (Velickovic et al., 2017). Both the fingerprint size and the attention range is chosen as 2-hop neighbors in our approach.

Results are reported in table 2. Our method has 2 variations, including ADSF-Nonparametric, where we learn a non-parametric decay profile w.r.t. node distance; ADSF-RWR, where we use random-walk with re-start to build fingerprints. For both cases, we have used 2-hop neighbors, and the restart probability is simply chosen as $c = 0.5$. As can be seen, our approach consistently improves the

| Methods | Cora | Citeseer | Pubmed |
|---|---|---|---|
| Gaussian Fields (Zhu et al., 2003) | 68.0% | 45.3% | 63.0% |
| Deep-Semi (Weston et al., 2012) | 59.0% | 59.6% | 71.7% |
| Manifold Reg. (Belkin et al., 2003) | 59.5% | 60.1% | 70.7% |
| Deep-Walk (Perozzi et al., 2014) | 67.2% | 43.2% | 65.3% |
| Link-based (Lu & Getoor, 2003) | 75.1% | 69.1% | 73.9% |
| Planetoid (Yang et al., 2016) | 75.7% | 64.7% | 77.2% |
| Chebyshev (Deffer rard et al., 2016) | 81.2% | 69.8% | 74.4% |
| GCN (Kipf & Welling et al., 2017) | 81.5% | 70.3% | 79.0% |
| Mixture-CNN ( Monti et al., 2016) | 81.7% | — | 79.0% |
| GAT (Velickovic et al., 2017) | 83.0±0.7% | 72.5±0.7% | 79.0±0.3% |
| ADSF-Nonparametric | 84.7±0.3% | 73.8±0.3% | 79.4±0.3% |
| ADSF-RWR | **85.4±0.3%** | **74.0±0.4%** | **81.2±0.3%** |

Table 2: Classification accuracy for different transductive methods on the benchmark data sets.

performance on all benchmark data sets. Since Velickovic et al. (2017) has performed an extensive set of evaluations in their work, we will use their reported results forbaseline methods. It is also worthwhile to note that our method only has a few more parameters compared with GAT (e.g., non-parametric decay profile $\mathbf{u}$, $c$ in RWR, and the mixing ratios in (6)). For example, for Cora data set, GAT model has about 91k parameters, while our method adds around 30 extra parameters on top of it (about 0.03% of the original model size).

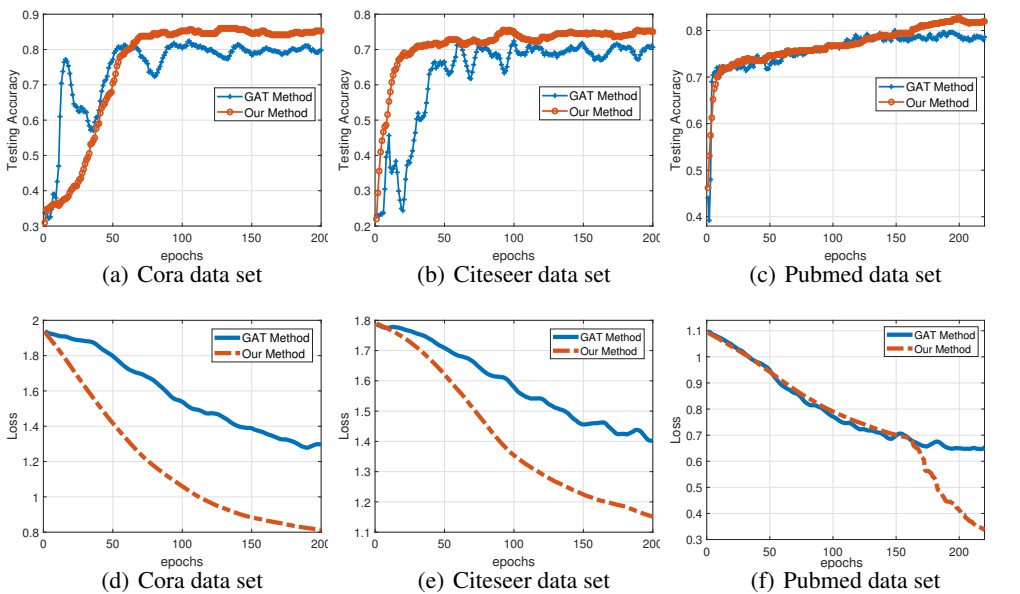

Figure 5: Convergence of the accuracy and loss of the proposed method and GAT method.

In Figure 5, we plot evolution of the testing accuracy and loss function (on the test split) for GAT and our method (ADSF-RWR). Since the loss functions of the two methods are the same, they are directly comparable. The accuracy of GAT fluctuates through iterations, while our approach is more stable and converges to a higher accuracy. The objective value of our method also converges faster thanks to the utilization of higher-order neighborhood information through structural fingerprints.

## 4.2 ABLATION STUDIES

In this section, We further examined some interesting details of the proposed method, all using the Cora data set and the RWR fingerprint construction scheme.

| Hop | $k' = 1$ | $k' = 2$ | $k' = 3$ |
|---|---|---|---|
| Cora | 83.0±0.7% | 15.2±6.8% | 13.3±4.7% |
| Citeseer | 72.5±0.7% | 68.4±4.1% | 65.7±4.5% |
| Pubmed | 79.0±0.3% | 78.2±0.5% | 67.2±0.3% |

Table 3: Performance of GAT under different choices of the neighborhood range.

**Impact of the fingerprint size**. Intuitively, the structural fingerprint should neither be too small or too large. We have two steps to control the effective size of the fingerprint. In constructing the fingerprint, we will first choose neighboring nodes within a certain range, namely the $k$-hop neighbors; then we will fine-tune the weight of each node in the fingerprint, either through nonparametric decay profile or the $c$ parameter in random walk. Here, for comparison, we have fixed $c = 0.5$ and varied $k$ as $k = 1, 2, 3$. As shown in Figure 6(a), the optimal order of neighborhood is two in this setting.

**Impact of the attention range**. In GAT, only direct neighbors have attention to each other, namely the attention range is $k' = 1$. In our approach, the optimal choice is $k' = 2$. The attention range determines the number of neighboring nodes involved in message passing. In the Cora dataset, the average 1st-order neighborhood size is $|\mathcal{N}_1| = \mathbf{3.9}$, while for 2nd-order neighbors $|\mathcal{N}_2| = \mathbf{42.5}$. Namely, our approach has involved 10 times more neighbors in message passing than GAT, yet the performance is improved instead. This is an evidence that our approach helps alleviate oversmoothing in GAT (or GNN's in general) by exploiting high-order structural details more effectively.

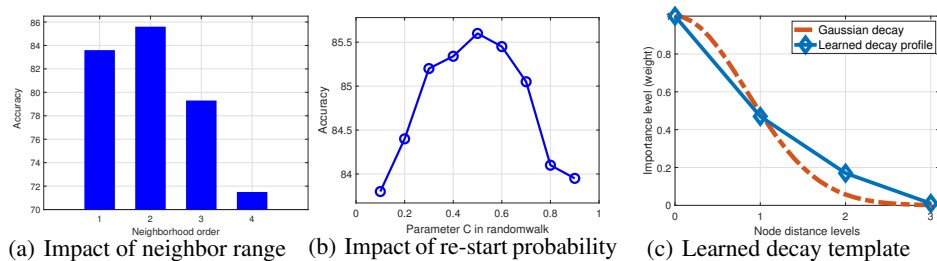

(a) Impact of neighbor range   (b) Impact of re-start probability   (c) Learned decay template

Figure 6: Detailed studies of the adaptive structural fingerprint method.

**Impact of the re-start probability in RWR**. In Figure 6(b), we plot the learning performance w.r.t. the choice of the $c$ parameter in RWR when fixing the neighborhood range $k = 2$. As can be seen, the best performance is around $c = 0.5$, meaning that the "random walking" and "restarting" should be given similar chances within 2-hop neighbors. Empirically, we can always use back-propagation to optimize the choice of $c$, which can be more adaptive to the choice of the $k$-hop neighbors.

**Non-parametric decay profile**. In Figure 6(c), we plot the non-parametric decay profile $\mathbf{u}$ learned by our method, when setting the neighborhood orders to $k = 3$. As can be seen, the first-order and second-order neighbors have higher weights, while the third-order neighbors almost have zero weights, meaning that they almost make no contributions to computing the structural attention. This is consistent to our evaluations in Figure 5(a), and demonstrates the power of our method in identifying useful high-order neighbors.

### 4.3 IMPACT OF NEIGHBORHOOD RANGE IN GAT

As suggested by the anonymous reviewer, we perform an empirical study on the GAT performance when larger attention range is considered. Note that in the original GAT by Velickovic et al. (2017), attention score is computed only for nodes that are direct neighbors with each other. Here, we increase the domain of interaction to up to 2-hop and 3-hop neighbors, with all other components remaining the same. As can be seen from Table 3, the performance of GAT is the best when choosing only 1-hop neighbors to compute attention scores, and deteriorates rapidly when larger neighborhood is considered.

We speculate that although larger attention range involves more neighbors and possibly more useful nodes, it bring a significant number of noisy nodes as well. As a result, the GAT performance is negatively affected. This seems to indicate that feature-based attention alone might not be sufficient in identifying noisy neighbors. In our approach, attention scores are computed from both node features and informative structural details, therefore it enables exploring a wider collection of higher-order-neighbors while simultaneously removing the impact of irrelevant nodes, validating the benefit of structure-based attention.

## 5 CONCLUSION AND FUTURE WORK

In this work, we proposed an adaptive structural fingerprint model to encode complex topological and structural information of the graph to improve graph representation learning. In the future, we will consider varying fingerprint parameters (e.g. decay profile) instead of sharing them across all the nodes; we will consider applying our approach to graph partitioning and community detection, where node features might be unavailable and graph structures will be the main source of information that can be explored; we will also extend our approach to the challenging problem of graph classification. On the theoretical side, we will borrow existing tools in semi-supervised learning and study the generalization performance of our approach on semi-supervised node embedding and classification.

## ACKNOWLEDGEMENT

We highly appreciate the valuable comments from anonymous reviewers, which allow us to significantly improve both the presentation and experimental design of our work. This work is supported in part by the National Science Foundation of China No.61672236, National Science Foundation of China No.61573107, Shanghai National Science Foundation 17ZR1444200, Shanghai Municipal Science and Technology Major Project No.2018SHZDZX01.

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
