# OpenReview forum: "Adaptive Structural Fingerprints for Graph Attention Networks"
_ICLR.cc/2020/Conference — Accept (Poster)_

### Official Review · AnonReviewer4 · 2019-10-17
**Official Blind Review #4**

**Rating:** 6

**Review:**

This work suggests a graph structure based methodology to augment the attention mechanism of graph neural networks.

The main idea is to explore the interaction  between different types of nodes of the local neighborhood of a root node. One component of this "fingerprinting" is a node weighting (closeness) that is computed using various methods proposed in the paper. The fingerprint is the vector of weights of all nodes in the local neighborhood.

For computing a closeness metric, the paper suggests random walks with restart, which has generally been used for graph clustering. Once the relative closeness to a node in the neighborhood is measured, the symmetric "structural interaction" between the fingerprint of two nodes is given by the Jaccard similarity of their structural fingerprints (or a smooth alternative thereof).  This structural similarity will be then considered in a (multi-head) attention mechanism using learned transfer functions.

The efficacy of the proposed method is tested on the Cora, Citeseer and Pubmed node classification benchmarks and compares favorably to non-augmented graph neural networks, beating all baselines on those datasets. Also the results in the paper beat those of the GResNet on Cora, but not on PubMed and , which is a recent paper not cited by this work.

In general this work goes into the direction of adding hand-engineered features to DeepLearning approaches. I am not a big fan of these methods, especially without significant theoratical justification. The approach is well motivated but very heuristical. Still the results presents a significant improvement of SOTA on those benchmark and the paper presents ideas  that seem to be generally useful for processing large-scale, structure-rich graph date. Hence I am in the favor of acceptance of this paper.

**Experience Assessment:**

I have published one or two papers in this area.

**Review Assessment: Checking Correctness Of Derivations And Theory:**

I assessed the sensibility of the derivations and theory.

**Review Assessment: Checking Correctness Of Experiments:**

I assessed the sensibility of the experiments.

**Review Assessment: Thoroughness In Paper Reading:**

I read the paper at least twice and used my best judgement in assessing the paper.

---

> ### Author Response · Authors · 2019-11-06
> **We thank the reviewer for the valuable comments and we have addressed your concerns.**
>
> First we highly appreciate the valuable comment from the reviewer and we are glad to have received positive comments, that our work “seem to be generally useful for processing large-scale, structure-rich graph data.”. The reviewer has given a great summary of main idea, and also valuable suggestions so that we can further improve the quality of our work. Our responses to the reviewer's comment is as follows.
>
>
> “The efficacy of the proposed method is tested on the Cora, Citeseer and Pubmed node classification benchmarks and compares favorably to non-augmented graph neural networks, beating all baselines on those datasets. Also the results in the paper beat those of the GResNet on Cora, but not on PubMed and , which is a recent paper not cited by this work.”
>
> --- We thank the reviewer for pointing out a related work on Residual Graph Attention Network, which also presents promising results on the data sets we have used. We have added this paper as a reference in our updated version. Using residual connection is an important way to improve the performance of graph attention networks in general.
>
> “In general this work goes into the direction of adding hand-engineered features to DeepLearning approaches.”
>
> ---We believe that a thorough theoretic analysis will add to the depth and quality of the paper. Along this direction, we are planning to borrow theoretic works in semi-supervised kernel learning, to see if the behavior of the proposed method can be analyzed more strictly. This may further shed light on how to improve the construction of the fingerprint from learning theoretic perspectives. We will add these discussion in the future work section considering that it involves a tremendous amount of effort and new contributions.

---

> > ### Comment · AnonReviewer4 · 2019-11-15
> > **Borderline**
> >
> > Thanks a lot for the updates. Still, my opinion stands.
> >
> > I find the paper noteworthy, but somewhat incremental, therefore my original recommendation stands: slightly favoring acceptance, but would not argue for its merits.

---

> > > ### Author Response · Authors · 2019-11-15
> > > **We highly appreciate the reviewer's decision**
> > >
> > > We would like to thank the reviewer again for the valuable time and comments.

---

### Official Review · AnonReviewer3 · 2019-10-17
**Official Blind Review #3**

**Rating:** 6

**Review:**

This paper extends the idea of self-attention in graph NNs, which is typically based on feature similarity between nodes, to include also structural similarity. This is done by computing for each node a value for each other node within its receptive field, calculated based on some distance metric from the node (either a Gaussian decay profile, or a learned weighting of number of hops distance, of based on fixed point of a random walk with restart). When evaluating the attention between two nodes, a function that compares the structural values between the two nodes (based on Jacard similarity) gives a score to the two nodes, which is further used for calculating the attention weight between them.

I found the idea to be elegant and well-explained, and overall the paper is well-written.
Using the structural similarity makes a lot of sense, and the proposed method is both easy to implement, and flexible — the structural similarity profile can be learned, which seems important for getting this idea to work in practice.

I wonder what happens if one uses only the structural similarity for the attention (without the feature similarity). Are there datasets where this would be sufficient? Even a toy task which is constructed such that the structure is more informative than the features could be a nice way to further demonstrate the idea of the paper.

The experiments show a clear yet small advantage to the proposed method over the conventional attention method (GAT).

Overall, this seems to be a solid contribution (even if the empirical results are a bit incremental) and I recommend acceptance.

**Experience Assessment:**

I do not know much about this area.

**Review Assessment: Checking Correctness Of Derivations And Theory:**

N/A

**Review Assessment: Checking Correctness Of Experiments:**

I assessed the sensibility of the experiments.

**Review Assessment: Thoroughness In Paper Reading:**

I read the paper at least twice and used my best judgement in assessing the paper.

---

> ### Author Response · Authors · 2019-11-06
> **We thank the reviewer for the valuable comments and we have addressed your concerns.**
>
> First we highly appreciate the valuable comment from the reviewer and we are glad to have received positive comments. The reviewer gives a great summarization of the main idea of our work, and also an interesting suggestion, which allows us to further improve the quality of our work. Our response is as follows.
>
> “I wonder what happens if one uses only the structural similarity for the attention (without the feature similarity). Are there datasets where this would be sufficient? Even a toy task which is constructed such that the structure is more informative than the features could be a nice way to further demonstrate the idea of the paper.”
>
> ---This is a very interesting idea to try, namely, only consider graph connections but without node content. It is related to graph partitioning or community detection, where the grouping is determined only using graph structure and topology (i.e., edge connections between nodes).  Examples include power grid, internet, and some biological networks, just to name a few. Indeed, we anticipate that our method will work very well on such data too. This is because our structural fingerprint for each node serves as a highly robust and informative structural identify descriptor, which can improve the evaluation of the similarities between two nodes. With better node-pair similarities, good clustering result is surely expected in graph partitioning.
>
> Besides, we can also tune the fingerprint size to achieve multi-scale clustering on graphs, which is an interesting future direction. We will start finding related graph data sets and experiment as the reviewer suggested, and to show the power of structural clues/information in graph clustering problems. We have included these discussions in the future work sectionn of our paper. We will also be pursuing these new directions and add results to the future arxiv versions of this paper.
>
> Finally, we would like to thank the reviewer again for your valuable time and comments.

---

### Official Review · AnonReviewer1 · 2019-10-23
**Official Blind Review #1**

**Rating:** 6

**Review:**

---- Problem setting and contribution summary ----
The paper considers the problem of graph node classification in a semi-supervised learning setting. When classifying a node, the decision is based on the node’s features, as well as by a weighted combination of its neighbors (where the weights are computed using a learnt attention mechanism). The authors extend upon the recent Graph Attention Networks (GAT) paper by proposing a different way of computing the attention over the neighboring nodes. This new attention mechanism takes into account not just their feature similarity, but also extra structure information, which also enables their method to attend not only over direct neighbors, but also up to k-hop neighbors.

---- Overall opinion ----
While I believe the general idea indeed has merit and empirically shows great promise, I believe the paper in its current state is not ready for publication. However, I believe that a more thorough revision can lead to an a publication with potential impact on the applications side.

---- Pros ----
1. The paper is easy to read.
2. I really appreciated the good visualizations (Figures 2,3,4) that indeed help in understanding the method.
3. Really good empirical results on the 3 datasets that were presented.

---- Major issues ----
1. Motivation:
I believe the paper is not well motivated from an applications perspective. In section 2.1., the authors did a good job explaining the limitations of current approaches on a generic graph structure, but this is only under the main assumption that a node should attend more to neighbors in its denser community than other neighbors that not connected so strongly (Fig 1). The issue that I have with this is that:
      a) Why should we take this assumption for granted? What are some concrete practical node classification problems where it is indeed better if a node attends to its neighbors in this way (as opposed to the GAT approach)?
      b) Even if the above is proven true, suppose node A in Fig. 1(a) attends with equal weights to all its 4 neighbors. That means node C (which is outside its densest community) gets 1/4, while the nodes inside the dense community get a total of ¾. That means node A puts most of its attention to the dense community anyway. In what conditions is it necessary to bias this attention even further?

2. Experiments:
While the reported accuracies for the three datasets look good and also the authors have provided a link to their code (great to see that!), I believe the experimental section is missing an important amount of details for reproducibility purposes and also for explaining how certain parameters have been chosen:
    a) There are no details on the model size and training procedure (hidden units, optimizer, learning rate schedule).
    b) Maybe I am missing this, but I don’t see any reference on what alpha and beta from equation (6) were used in the experiments.
    c) What is the value of k in Table 2? How did you choose it? I believe Figure 5 shows test accuracies for different k, but I hope the authors did not choose k based on the test set performance.
    d) How did you select the structural fingerprint to be 3?
     e) “...optimizing c through the learning process also gives very similar choice” → how did you optimize c exactly?
     f) Fig 5 a) Why does increasing the number of hops to 3 or 4 decrease the performance so much? Shouldn’t the attention weights learn to ignore the further neighbors, if they are not useful?

	Another important question regarding experiments: since the ablation study shows that the optimal neighbor range is actually 2, a natural baseline to compare with would be something similar to GAT, where an attention weight is applied to all neighbors within two hops (basically skip the fingerprint step, and assume s_{ij} is 1 for all neighbors within 2 hops, and 0 otherwise).

	Also regarding experiments, these 3 datasets, although common across many recent graph node classification papers, they are known to be quite limited (small in size and not very representative of real world). Since GAT is your main competitor, why not also show experiments on the PPI dataset they also test?

3. Writing quality:
While the language is clear and easy to follow, there are many grammatical mistakes throughout the paper (e.g. “benefitial”, subject-verb agreement).

---- More minor issues ----
    a) Section 2.1: One could argue that GAT also contains longer range node dependencies through the node embeddings it learns. Since the node embeddings is trained through gradient descent, and at each iteration the embedding of a node changes according to its neighbors, you could say that information does get propagated from the neighbor’s neighbors.
     b) Why do you need a LeakyRelu in Equation (5) ? Also, aren’t e_{ij} non-negative anyway (in which case LeakyRelu doesn’t change anything)?
     c) Why would a Sigmoid be a good choice for alpha and beta in Eq. (6)?
     d) Section 3: A bag of words is typically represented as a binary vector, which is also categorical.
     e) Please use \citet when specifically referring to the authors of a paper as part of your sentence (e.g. “Following Velickovic et. al., 2017 we….” as opposed to “Following (Velickovic et al., 2017), we....”).


**Experience Assessment:**

I have published one or two papers in this area.

**Review Assessment: Checking Correctness Of Derivations And Theory:**

N/A

**Review Assessment: Checking Correctness Of Experiments:**

I carefully checked the experiments.

**Review Assessment: Thoroughness In Paper Reading:**

I read the paper thoroughly.

---

> ### Author Response · Authors · 2019-11-06
> **We thank the reviewer for the valuable comments and we have addressed your concerns.**
>
> We feel the reviewer has read our paper in great depth, which is already a reward to our work. We're greatly inspired by the valuable comments, which allow us to better elaborate our idea and obtain interesting new experimental findings.
>
> 1.Motivation:
>
> “I believe the paper is not well motivated from an applications perspective. .. under the main assumption that a node should attend more to neighbors in its denser community than other neighbors that not connected so strongly”
>
> ---First we want to clarify a subtle difference between our approach and reviewer’s comment. What reviewer summarizes “a node should attend more…” is not on computing the attention yet, but rather on the initial step of building fingerprint (receptive domain) of each node. So, allow us to rephrase to “the structural fingerprint of a node should emphasize more on densely inter-connected neighbors within predefined vicinity (shown in Fig3).
>
> ---*Validity*: similar assumption is prevalent in many learning paradigms: min-cut/normalized-cut minimizes cross connections between two clusters/subgraphs [Shi and Malik, PAMI 00]; low-density-separation assumes cluster boundaries pass through low-density regions, leading to successful semi-supervised clustering [Chapelle and Zien AI&STAT 05]; mean-shift algorithm explicitly finds local density peaks as clusters [Comaniciu and Meer, PAMI 02]. These works are cited tens of thousands of times, reflecting wide acceptance of the assumption that “structural” clues such as dense-subgraphs/high-density-regions are valuable for clustering.
>
> ---*Consequence*: due to this assumption, fingerprint (receptive field) of each node can be constructed more accurately. This is because neighbor nodes in higher-density regions, which are more likely from the same cluster (by references above), will have higher weights in our fingerprint. So, structural relation (attention) between two nodes can be estimated more accurately, leading to better accuracy.
>
> ---Finally, we also proposed Gaussian-decay to build fingerprint. It makes no assumption but ''closer nodes are more important''. It effectively improves GAT too. Clearly, key to both success is the use of structural information to improve depiction of a node.
>
> We added these in Sec2.1 (last 2 paragraphs); Sec2.3 (last paragraph)
>
>
> “a) Why should we take this assumption for granted…”
>
> ---The assumption has been used widely in clustering/semi-supervised learning. Besides, we did NOT directly enforce it in attention and require "one node has to attend high-density neighbors more", instead, we used it to build more accurate fingerprints, which then improves accuracy of subsequent attention computation (which is a bit subtle).
>
> "b) ...suppose node A in Fig. 1(a)...?"
>
> ---This pic was meant to show node distance may not reflect true closeness good enough. If one wants to link it to attention, one can put it this way: fingerprint of A and B emphasize more on left subgraph; that for C will emphasize right subgraph. So, structure similarity between A and C is small, while for A and B is large.
>
>
>
> 2. Experiments
>
> “a) There are no details on the model size”
>
> ---Our setting is the same as GAT; we added detailes now in sec4.1.
>
> “ b) …reference on alpha and beta from eq (6)”.
>
> ---we used scalar alpha/beta.
>
> “ c) What is the value of k in Table2?”
>
> --- K = 3. It is not chosen by looking into Fig5a (there, best K is 2)
>
> “d) How did you select the structural fingerprint to be 3?”
>
> ---Number of K-hop neighbors increases quickly with K. For efficiency, we restrict K to 3;  larger K gives more information but more costly.
>
> “e) how did you optimize c”?
>
> ---Back-propagation
>
> “f) Fig 5 a) Why does increasing number of hops to 3 or 4 decrease …Shouldn’t attention weights learn to ignore further neighbors…”
>
> ---Our method surely learns when to ignore far-away neighbors. But in this ablation study, we varied K while Fixing c=0.5, the parameter controlling effective size of fingerprint, as stated in original submission (last line, page 7). In final experiments (Table 2), we optimized c anyway.
>
> “a natural baseline...would be GAT applied to all neighbors within two…”
>
> ---Great suggestion! We have tried GAT with 2/3-hop neighbors; its performance drops.  Namely, GAT prefers small neighbors (1-hop neighbors). This reslt shows that our improvement is not just due to large neighbors, but rather the useful structural clues extracted in our method. We added these in the supplementary section.
>
> “why not also show experiments on PPI dataset”
>
> ---Preliminary one round result is promising (>97% accuracy). Yet, due to large data size and limited computing resource, we cannot finish 10 repeated runs on time (to compute average performance), so we cannot include this incomplete result. We will add them in future arxiv versions.
>
>
>
> 3. Minor comments
>
> ---we have revised accordingly, and  proof-read the paper and corrected typos. We highly appreciate the reviewer for the detailed checking.

---

> > ### Author Response · Authors · 2019-11-15
> > **Brief summary of our revisions**
> >
> > For reviewer's convenience, we asummarize our revisions as follows
> >
> > 1. Adding more references supporting the assumption that ''dense-connections/high-density regisions'' are valuable structural clues for clustering (normalized cut, low-density separation, and mean shift algorithm); related discussions are added to the last 2 paragraphs of Sec 2.1, and last paragraph of Sec 2.3.
> >
> > 2. We have added a new paragraph in sec4.1 (2nd paragraph) describing the detailed model setting, which is basically the same as in the GAT framework, as well as the choice of the neighborhood size
> >
> > 3. We have added a section in the supplementary material, studying the performance of the GAT method when larger neighborhood is used (i.e., 2-hop neighbors), as suggested by the reviewer.
> >
> > 4. We have clarified the discussion of the ablation study in Sec 4.2, in particular for Fig6(a), namely, the results here are obtained when fixing c, the restart probability, to be 0.5, in order to better study the impact of k (neighbor-degree).
> >
> > 5. We have thoroughly proofread the paper and corrected typos and format of references

---

> > ### Comment · AnonReviewer1 · 2019-11-16
> > **Updated review score**
> >
> > I thank the authors for addressing my comments. I particularly appreciate the details and experiments added to the paper.  In light of these changes, I have updated my score.

---

### Decision · Program_Chairs · 2019-12-19

**Decision:**

Accept (Poster)

**Comment:**

This paper is consistently supported by all three reviewers during initial review and discussions. Thus an accept is recommended.